# Human immunocompetent choroid-on-chip: a novel tool for studying ocular effects of biological drugs

Madalena Cipriano [1,2], Katharina Schlünder[1,3], Christopher Probst[4], Kirstin Linke[4], Martin Weiss [3,5], Mona Julia Fischer [1], Lena Mesch[5,6], Kevin Achberger[6], Stefan Liebau[6], Marina Mesquida[7], Valeria Nicolini[7], Anneliese Schneider[8], Anna Maria Giusti[8], Stefan Kustermann [7✉] & Peter Loskill [1,2,3✉]

Disorders of the eye leading to visual impairment are a major issue that affects millions of people. On the other side ocular toxicities were described for e.g. molecularly targeted therapies in oncology and may hamper their development. Current ocular model systems feature a number of limitations affecting human-relevance and availability. To find new options for pharmacological treatment and assess mechanisms of toxicity, hence, novel complex model systems that are human-relevant and readily available are urgently required. Here, we report the development of a human immunocompetent Choroid-on-Chip (CoC), a human cell-based in vitro model of the choroid layer of the eye integrating melanocytes and microvascular endothelial cells, covered by a layer of retinal pigmented epithelial cells. Immunocompetence is achieved by perfusion of peripheral immune cells. We demonstrate controlled immune cell recruitment into the stromal compartments through a vascular monolayer and in vivo-like cytokine release profiles. To investigate applicability for both efficacy testing of immunosuppressive compounds as well as safety profiling of immunoactivating antibodies, we exposed the CoCs to cyclosporine and tested CD3 bispecific antibodies.

[1] Institute for Biomedical Engineering, Eberhard Karls University Tübingen, Tübingen, Germany. [2] 3R-Center for In vitro Models and Alternatives to Animal Testing, Eberhard Karls University Tübingen, Tübingen, Germany. [3] NMI Natural and Medical Sciences Institute at the University of Tübingen, Reutlingen, Germany. [4] Fraunhofer Institute for Interfacial Engineering and Biotechnology IGB, Stuttgart, Germany. [5] Department of Women's Health, Research Institute for Women's Health, Eberhard Karls University Tübingen, Tübingen, Germany. [6] Institute of Neuroanatomy & Developmental Biology (INDB), Eberhard Karls University Tübingen, Tübingen, Germany. [7] Pharmaceutical Sciences, Roche Pharma Research and Early Development, Roche Innovation Center Basel, F. Hoffmann-La Roche Ltd., Basel, Switzerland. [8] Pharmaceutical Sciences, Roche Pharma Research and Early Development, Roche Innovation Center Zurich, F. Hoffmann-La Roche Ltd., Basel, Switzerland. ✉email: stefan.kustermann@roche.com; peter.loskill@uni-tuebingen.de

Organ-on-chip (OoC) technology allows building complex in vitro models tailored specifically to the tissue/organ needs. OoC models mimic the microphysiological environment cells experience in a tissue including the vasculature-like perfusion. They are developed to potentiate several functional readouts using very low cell numbers. Over the past years, the technology has emerged as a powerful tool to support drug discovery and development with a potential for pharmaceutical R&D cost reduction[1,2]. The rise of complex treatment modalities, increasing attrition rates and low of predictivity of current model systems created an urgent need for human-relevant and well-characterized in vitro models to support drug development (i) in efficacy testing by building in vitro disease models and (ii) in toxicity testing by providing a unique tool for mechanistic studies. Of particular interest for both efficacy and toxicity testing, is the human eye. Millions of people worldwide are affected by ocular disorders leading to visual impairment[3]. Moreover, ocular toxicities have emerged as an issue of targeted therapies[4] and have been reported for immunotherapies[5]. Hence, to develop ophthalmic drugs and study ocular toxicity, novel human-relevant ocular tissue models are urgently needed[6].

One such ocular tissue, is the choroid of the eye which belongs to the uveal tract. It is a thin, highly vascularized and pigmented tissue positioned between the neurosensory retina and the sclera, which takes care of the outer retinal demands for nutrition supply and removal of toxic metabolic products. The human choroidal tissue is underlined by the retinal pigmented epithelium (RPE) and its main cell types are melanocytes, choriocapillary microvascular endothelial cells (MVEC), fibrocytes and immune cells[7]. Uveitis refers to a group of ocular disorders characterized by inflammation of the uveal tract, which encompasses the iris, ciliary body and choroid. The inflammatory state is characterized by cytokine release and substantial T cell recruitment and infiltration[8]. Uveitis is the 4th leading cause of blindness, responsible for 10% of blindness in the US. Its origin can be infectious (20%) or noninfectious (immune-mediated, drug induced or idiopathic)[9]. Uveitis is categorized based on the primary anatomical location of inflammation. Posterior uveitis involves inflammation of the retina, choroid and sometimes the optic nerve and in spite of not being the most common type of uveitis it is indeed the most sight-threatening kind, with potential for causing severe structural damage and visual impairment[10]. Immunotherapies have been responsible for an increase of drug-induced uveitis and ocular side effects are observed in up to 70% of patients under immunoncological treatments including inhibitors of VEGFR, EGFR, tumor specific proteins, as well as estrogen receptor modulators, interferons[11] and checkpoint inhibitors[12,13]. All types of checkpoint inhibitors have shown ocular side effects. Uveal effusion was reported for anti-PD-1/PD-L1 checkpoint inhibitor agents, both small molecules and monoclonal antibodies[14]. Ipilimumab, a monoclonal antibody targeting cytotoxic T lymphocyte antigen-4 (CTLA-4) has been associated with choroidal neovascularization that leads to the disruption of the choroid-retina barrier causing meaningful visual impairment. Interestingly, uveitis treatment covers immunosuppression with small molecules (e.g., corticosteroids, cyclosporine, tacrolimus, or methotrexate) and more recently biologic therapies targeting the inhibition of activated T cells VEGF (e.g., adalimumab, infliximab)[8] as well as pro-inflammatory cytokines such as TNF-α and IL-6 (e.g., tocilizumab and sarilumab in non-infectious intermediate, posterior, and panuveitis)[15].

The lack of cross-reactive, pharmacologically relevant animal species poses a challenge in nonclinical safety assessment in the development of novel complex treatment modalities as in cancer immunotherapy. Translation of findings e.g., in non-human primates to patients can be hampered due to species differences in their immune systems[16,17]. Two remarkable differences between humans and rhesus and cynomolgus macaques are that CD4 + CD8 + double-positive T cells are more abundant in macaques and display unique responses to different type I interferons and that cynomolgus plasmacytoid dendritic cells (pDCs) have a unique pSTAT5 response to IL-6[18]. This is particularly interesting for posterior uveitis since IL-6 is a therapeutic target under phase I/II clinical investigation[15] and pDCs have been reported to be decreased relative to monocyte derived DCs possibly leading to biased CD4⁺ T cell immune responses (increased Th1 and decreased Treg cells) in non-infectious uveitis in humans[19]. Hence, advanced human-based and immunocompetent in vitro models of the choroid of the eye are urgently required to study the interplay between the vascular network and the human immune system. Such a model could support research on early detection of drug-induced uveitis and on the discovery of new therapeutic strategies for diseases such as posterior uveitis, regardless of its etiology. Expended testing with several donors would further help to inform about the mechanistic insights of idiosyncratic effects in a patient-specific manner.

The few available in vitro models of the human choroid are mostly 2D, integrate few cell types and lack vascularization or immune components; aspects that are all crucial for mechanisms of toxicity or disease triggering events[20–22]. In general, in vitro models that study immune cell infiltration are still limited and encompass mostly three-dimensional (3D) skin[23] and tumor models[24–27].

To address the key physiological characteristics related to choroidal drug reactions and considering immunology, we developed a human immunocompetent choroidal in vitro model that mimics the tissue vascularization, pigmentation and immune response in the presence of circulating immune cells. Here we provide a comprehensive characterization of the novel model and describe how the immunocompetent chip responded (i) to immunosuppressive treatment with cyclosporine upon T cell activation and (ii) to a T cell bispecific antibody (TCB) containing the T cell receptor binding domain.

## Results and discussion

### Design, concept and characterization of the Choroid-on-Chip

*Platform concept.* To mimic the tissue complexity of the human choroid, we designed a 3-channel microfluidic platform that allows for the perfusion of circulating immune cells. The Choroid-on-Chip (CoC) comprises three main cellular components (Fig. 1a–c). The epithelium consists of a monolayer of human induced pluripotent stem cell (iPSC)-derived RPE in the top channel. The endothelium consists of two confluent monolayers of human primary MVECs seeded in the central channel facing on the upper side the RPE and on the lower side the melanocyte compartment. This channel is perfused using a syringe pump at a flow rate of 40 μL/h. The stromal component in the bottom channel features a cell-laden hydrogel incorporating melanocytes at defined cell densities in a 3D arrangement (Fig. 1d, orthogonal view). All the epithelial, endothelial and stromal components together form a robust, viable 3D tissue (Fig. 1d). The cells are homogenously seeded and distributed in a pigmented tissue throughout the entire length of the chip (Fig. 1e). The dimensions of the chip were chosen to match in vivo tissue dimensions: The total height of the 3D tissue on chip is 300 μm; the human choroidal thickness was estimated to be $266.8 \pm 78.0$ μm by Yiu et al.[28].

*Outer blood-retina barrier characterization.* To mimic the outer blood-retina barrier (oBRB), RPE and MVECs were cultured on

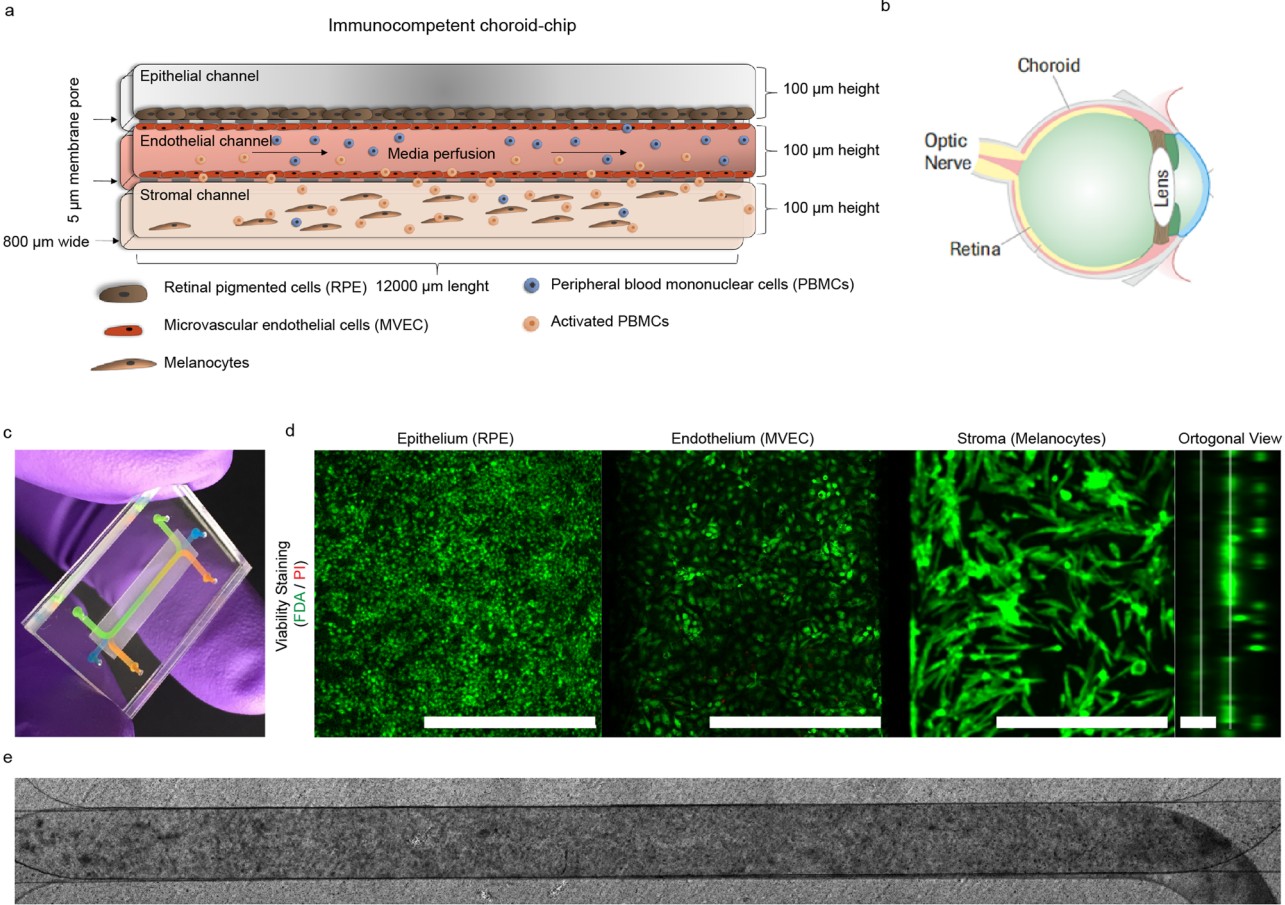

**Fig. 1 Concept of the CoC model. a** Schematic representation of the different cell types within the chip. **b** Schematic of the layers of the eye. **c** CoC platform filled with watercolor demonstrating the location of the RPE (green), the endothelial (orange) and melanocyte (blue) channels. **d** Live/Dead staining performed with Fluorescein diacetate/Propidium Iodide (FDA/PI) to demonstrate the viability of the long-term culture (2 weeks) of the CoC as well an orthogonal view. Scale bar = 500 μm for planar images and 100 μm for the orthogonal view. **e** Bright field image of the whole chip after 2 weeks in culture and 24 h if immune cell perfusion.

the two sides of the upper membrane, top and bottom respectively (Fig. 1a). The membranes were coated with Laminin, Collagen and Fibronectin, as described in detail in the Methods section, to mimic the Bruchs´ Membrane composition[29]. The RPE formed a pigmented monolayer throughout the entire chip expressing tight junction protein zonula occludens 1 (ZO-1) and tyrosinase-related protein 1 (TYRP1), involved in the generation of melanin[30] (Fig. 2a). MVECs formed confluent and CD31 positive monolayers throughout the entire length of the endothelial channel (Fig. 2b); endothelial layers were negative for ZO-1 in the CoC. To further assess how our model mimics the oBRB, we evaluated the permeability of the endothelial layer towards the RPE side (oBRB) and towards the stromal side of the tissue (Fig. 2c). Our data showed retention of both Carboxyfluorescein (0.377 kDa) and Dextran Texas Red (70 kDa) in the presence of the cell monolayers in all conditions. The retention was higher for the oBRB than for the endothelial-stromal barrier (twofold) and for the larger molecular weight molecule compared to the lower one.

A comprehensive quantitative analysis of the absolute difference between the permeability of the endothelial-stromal barrier and the oBRB is hampered by the differences in tissue composition of RPE and stromal channel: the hydrogel in the stromal channel leads to a slower intra-channel diffusion in contrast to the medium in the RPE compartment. However, it is worth mentioning that the relative differences are indeed in

alignment with the in vivo situation and ex-vivo data[31] showing a tighter barrier where the RPE is located. The permeability of the choriocapillaris is expected to be 1–2 orders of magnitude higher than the oBRB[32]. The endothelial-RPE barrier is reported to be 30 times less permeable to the 70 kDa dextran than to the carboxyfluorescein[31]. We did not observe the same magnitude of differences in the permeability as described in ex-vivo and other in vitro studies. However, the chip was not specifically designed to quantify barrier properties but to study effects of drugs and/or immune response in the stromal and epithelial compartments.

*Choroidal stroma and melanocyte characterization.* The melanocyte seeding process in the CoC was designed to achieve two different melanocyte densities, 5 and 50% (Fig. 3a, b). Morphology, pigmentation, proliferation (Ki-67) and viability were characterized for high and low melanocyte cell densities after 2 weeks in culture (Fig. 3a, b, respectively). Three-dimensional cell distribution and active production of melanin were confirmed by melanin autofluorescence imaging and by the presence of TYRP1 in these cells (Fig. 3c). Interestingly, the main functional difference between the chips cultured with high and low melanocyte density was in IL-6, secreted proportionally to the melanocyte cell density, conversely to secretion of the pro-inflammatory cytokine TNF-α (Fig. 3d).

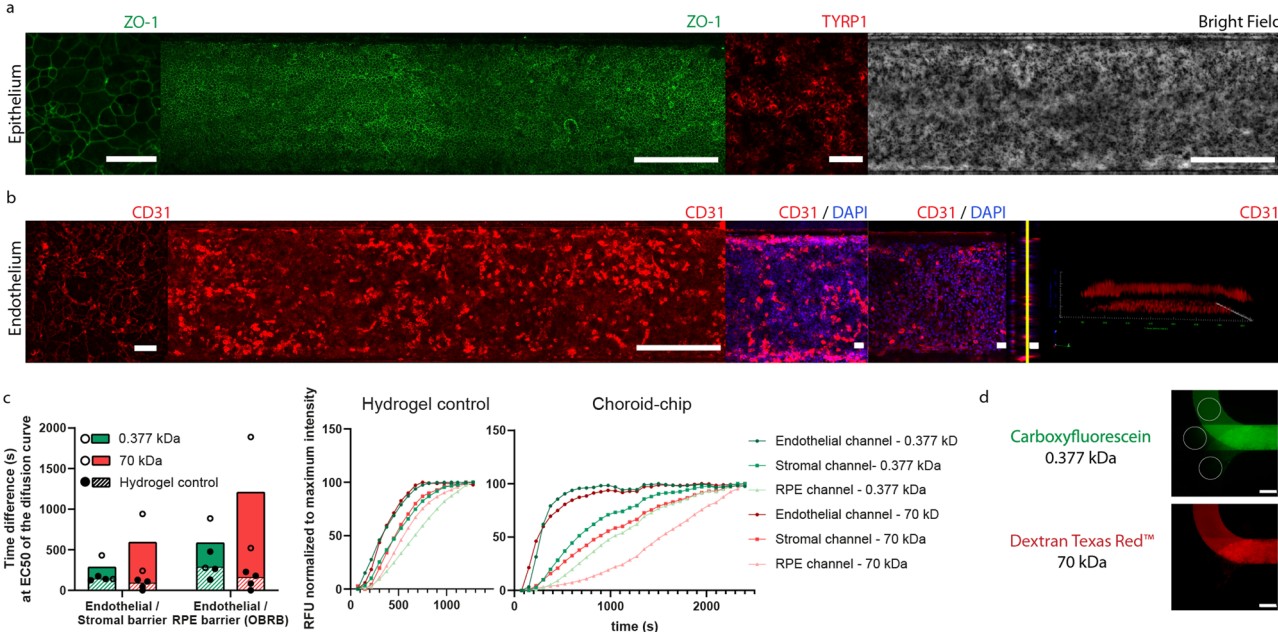

**Fig. 2 Endothelial and outer blood-retina barrier (oBRB) characterization after 2 weeks in culture. a** Immunofluorescence staining demonstrating the phenotype and structure of the retinal pigmented epithelium as a confluent monolayer expressing the tight junction marker zonula occludens (ZO-1, in green) at high magnification (left) and throughout the whole cell culture surface on the chip (middle, long image of ZO-1, in green) as well as melanin production, demonstrated by the expression of TYRP1 (red) and pigmentation demonstrated in the bright field imaging (right, gray). **b** Immunofluorescence staining demonstrating the phenotype and structure of the endothelium formed by microvascular endothelial cells positive for CD31 (red), presenting a stronger staining in the cell junctions when cultured in the absence of RPE and melanocytes (left) than in co-culture (middle, long image of CD31, in red). The DAPI/CD31 staining represents the bottom endothelial layer (left), top endothelial layer (middle) and orthogonal side view of both, with a yellow line corresponding to the bottom membrane. The 3D render (right) confirmed the presence of 2 confluent layers. (**c**) Quantification of the relative permeability of the endothelial (MVEC)/stromal and oBRB barriers when perfused with low (Carboxyfluorescein, 0.377 kDa) and high molecular weight (Dextran Texas Red, 70 kDa) fluorescence molecules. Relative permeability is presented as a measure of the lag time difference to achieve 50% of the compound concentration (right) extracted from the molecule diffusion kinetic curve ($n = 2$ for the CoC and $n = 3$ for the Hydrogel control, bars represent SEM) (left) and a representation of the kinetic profile measured for one chip, normalized to the highest fluorescence intensity in each channel of the CoC and for both molecular weight molecules. **d** Representative images of the fluorescent measurements, labeled with white circles, performed to quantify relative permeability. The measurements were performed in each of the individual channel, where there is no channel overlap, on the inlet side of the chip. The top circle corresponds to the MVEC channel, the middle to the stromal channel and the bottom to the RPE channel. The images show a fluorophore filled MVEC channel after 552 s of perfusion. Scale bars are 500 μm, except for square images and the orthogonal view in in (**a**) and (**b**), where the scale bars represent 50 μm.

In vivo, uveal melanocytes are located in the choroidal stroma and are essential for the normal ocular homeostasis and function, namely for light absorption, regulation of oxidative stress, immune regulation, angiogenesis and inflammation[33]. Uveal melanocyte size and pigmentation varies within ethnicities and species. Rhesus macaques feature 50% of the choroidal tissue filled with melanocytes in opposition to 5% observed for white humans, corresponding to a density of 34.2 uveal melanocytes per 10.000 μm² with an average size of 28.5 μm[2,28]. The CoC aimed at recapitulating those cell densities with the low and high melanocyte densities, respectively, being able to address possible species differences for non-clinical drug development. Different melanocyte densities are accompanied by differences in melanin content with pharmacological implications such as binding of small molecule drugs to melanin[34] and by different IL-6 levels. IL-6 is secreted by both melanocytes and RPE and is a key cytokine for the immunological state of the eye involved in maintaining uveal tissue homeostasis[35]. IL-6 also plays a central role in ocular disease: intraocular IL-6 levels have been found elevated in a plethora of retinal diseases including uveitis, AMD and diabetic eye disease[36,37]. Interestingly, IL-6 and TNF-α were the only cytokines detected in human vitreous of controls[37], which corresponds to the cytokine profile of the low melanocyte

density condition of our model (Fig. 3c). This condition was, hence, used for the further studies.

*Perfusion of circulating immune cells.* To integrate an immune component into the CoC, freshly isolated human peripheral blood mononuclear cells (PBMCs) were perfused through the endothelial channel. The perfusion process and parameters were carefully chosen to assure cell viability while avoiding immune cell activation that would lead to an uncontrolled and undesired immune cell response (Fig. 4). The capacity of the immune cells to adhere to the membrane and migrate towards the melanocyte compartment was initially assessed in acellular chips. We perfused PBMCs and activated T cells ("Act", treated with anti CD3/CD28 antigens) through chips containing either bare hydrogel or hydrogel supplemented with CCL19, a chemokine responsible for directing T cells from the blood stream into lymph nodes[38]. In the absence of T cell activation and CCL19, almost no immune cells were present in the melanocyte compartment whereas cell recruitment was clearly observable in the presence of CCL19 and T cell activation (Fig. 4a).

In the second step, we perfused PBMCs and activated T cells through the fully assembled CoC to better understand the immune cell recruitment into the melanocyte compartment and evaluate if T cell activation would mimic the disease phenotype of

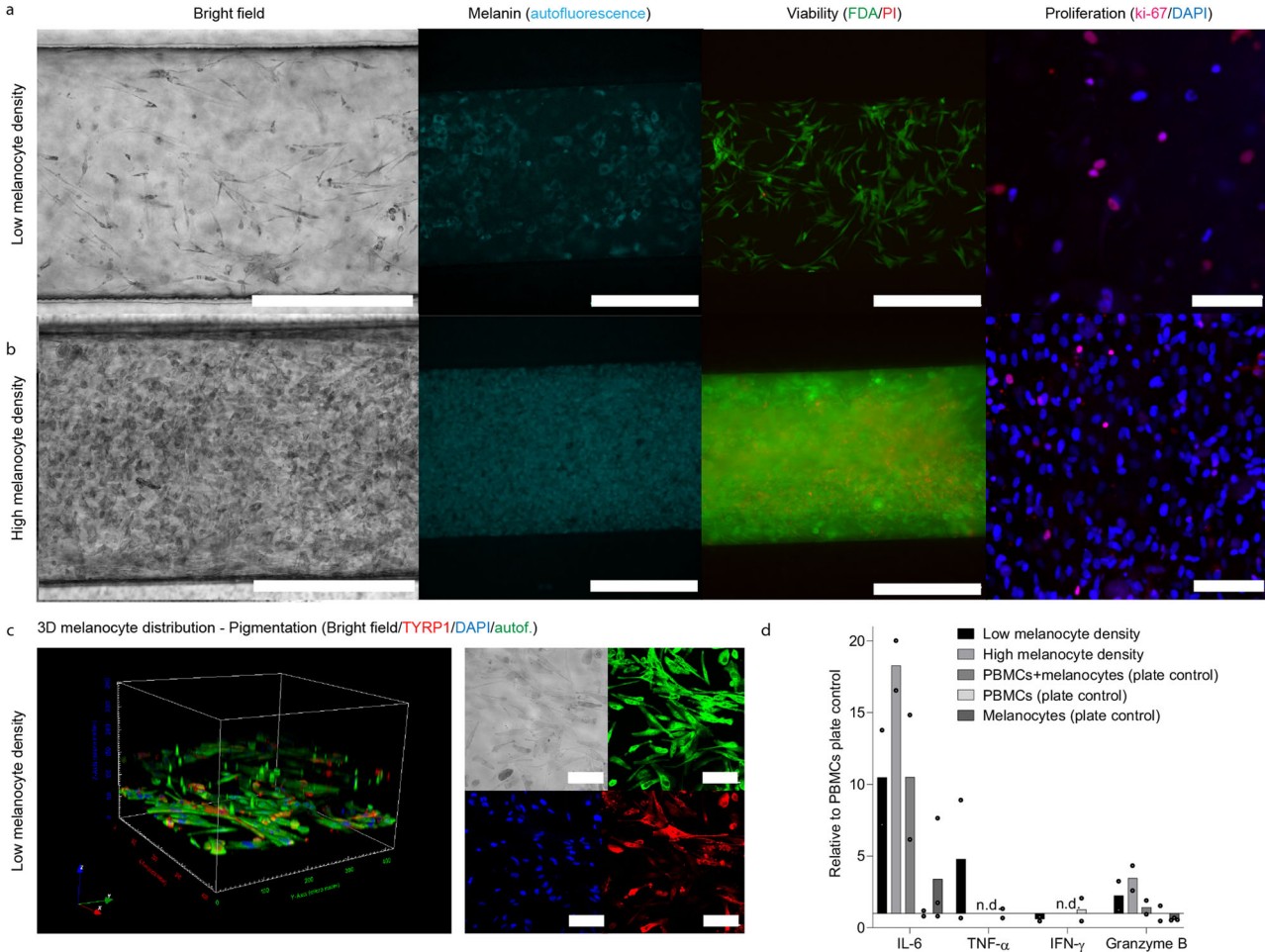

**Fig. 3 Characterization of the melanocyte compartment after 2 weeks in culture.** Morphology of pigmented melanocytes seeded at low (**a**, top) and high (**b**, bottom) densities and embedded in the dextran hydrogel containing Arginylglycylaspartic acid (RGD peptide): Bright field images demonstrate pigmentation; autofluorescence shows melanin, FDA/PI labeling indicates viability (Epifluorescence imaging of the whole chip), and KI-67 staining (Confocal imaging) a low proliferative state. Scale bar: 500 μm (Ki-67 images: 100 μm); (**c**) 3D representation of the cell distribution in the melanocyte compartment (left). Immunofluorescence microscopy highlighting pigmentation (bright filed; top left), melanin autofluorescence (top right), cell nuclei (DAPI; bottom left) and the presence of TYRP1 (bottom right) (Confocal imaging). **d** Effect of melanocyte density on the cytokine secretion in the presence of peripheral blood mononuclear cells (PBMCs). The bars represent the average values and the dots the individual values (*n* = independent biological replicates). Statistical analysis is detailed in Supplementary data 1, including individual data points.

uveitis. After 24 h of linear perfusion, more than 50% of the perfused immune cells remained in the chip (Fig. 4b); moreover, cells in the effluent were not proliferative (<5% of Ki-67 positive cells, Supplementary Fig. 1). The cell viability remained above 75% for both PBMCs and activated T cells. Fraction of T cells positive for activation marker CD69 was twofold lower in the chip effluent than for plate controls (Fig. 4c). In contrast to this, CD25, a later marker of T cell activation was not changed by activation, probably due to the short period of activation of 24 h[39]. This suggests that activated cells stay in the chip, adhering to the endothelium and/or migrating towards the other compartments as also observed by quantitative image analysis of the entire chip and measurement of cytokines in the chip effluent. Treatment with CD3/CD28 antigens led to an increase in PBMC and T cell migration particularly towards the lower half of the melanocyte compartment (Fig. 4d, e). T cell activation resulted in consistently higher cytokine levels for all tested cytokines. IL-10, particularly was not detected in PBMCs and detected at a low concentration in activated cells, consistent with the early time point of 24 h (Fig. 4f). IL-6 mediates T cell migration in the presence of extracellular matrix, requiring

integrin signal transduction pathways and a gradient for chemotactic migration[40]. Therefore, IL-6 increase could explain the increased recruitment of immune cells. This is reinforced by the higher proportion of immune cells that migrated into the melanocyte compartment in high melanocyte density chips: 3-fold increase for all immune cells and 7-fold for T cells (Supplementary Fig. 2), when perfused with PBMCs. In high melanocyte density chips, T cell activation did not result in an increase of immune cell recruitment (Supplementary Fig. 3).

## Exposure of the immunocompetent CoC to immunomodulatory drugs

*Cyclosporine A.* The capacity of the CoC to respond specifically to immunomodulatory stimuli was evaluated using the well-known immunosuppressor Cyclosporine A (CsA). The CoCs were exposed to different doses of CsA while perfused with PBMCs activated via anti CD3/CD28 antigens. CsA treatment did not significantly affect immune cell viability (Fig. 5A). However, the cell recovery in the effluents was significantly higher for CsA treated CoCs than for controls with activated T cells or PBMCs (Fig. 5b). The immunosuppressing effect of CsA was observed in

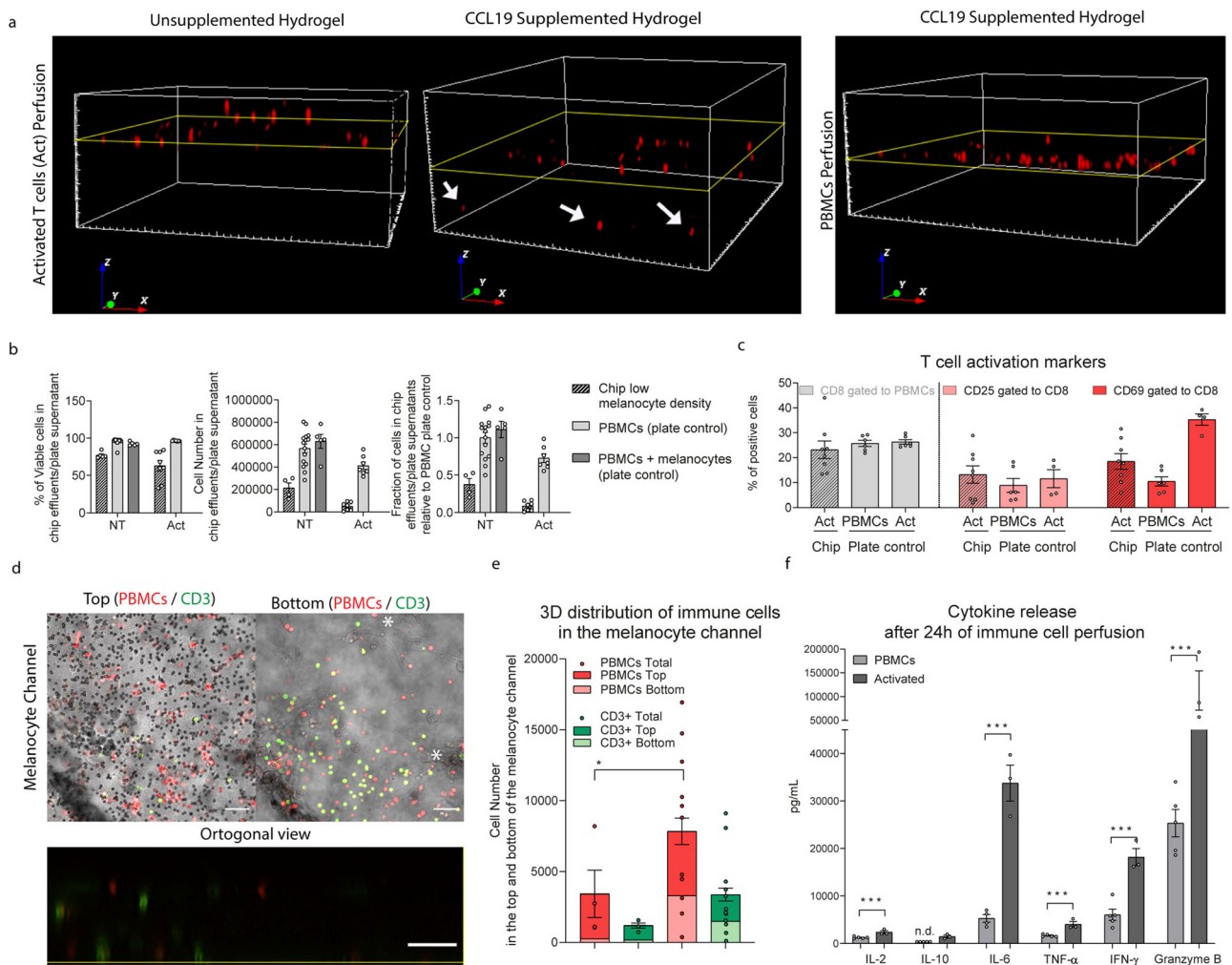

**Fig. 4 Validation of PBMC perfusion, viability, recruitment and cytokine release upon activation with anti CD3/CD28 antigens (Act). a** 3D representations of the evaluation of immune cell recruitment on chip containing only hydrogel (left) or supplemented with CCL19 (middle and right) and perfused with Deep Red cell tracker labeled PBMCs, treated with anti CD3/CD28 antigens (middle and left) or non-treated (right) showing that in the absence of a stimuli, there is no immune cell recruitment. The yellow line represents the position of the bottom membrane of the chip. **b** Evaluation of the cells collected after 24 h of perfusion on its viability (left), concentration (middle) and fraction of recovery relative to the plate control ($n = 4$–15) measured by flow cytometry. The effluents were collected from the chips with low melanocyte density (black) and from control well plates with (dark gray) or without co-culture with melanocytes (light gray). **c** Quantification of CD8 positive cells and of T cell activation markers CD25 and CD69 within the CD8 positive population by flow cytometry analysis of the effluents in the chips with low melanocyte density (striped bars) and on the plate controls ($n = 4$–8). **d** Evaluation of the immune cell recruitment towards the 3D melanocyte compartment in chips with low melanocyte density and activated (Act) distinguishing PBMCs (red) and T cells (green, labeled with an anti-CD3 antibody) within the melanocytes (*). The figure shows the top of the compartment where the chip bottom membrane is located and its pores are focused (right) and the bottom of the compartment (left), with an orthogonal representation of the whole chip height (bottom). Scale bar is 50 μm. **e** Quantification of immune cell recruitment by image analysis of the whole chip with low melanocyte density, distinguishing PBMCs (cell tracker labeled prior perfusion) and T cells (CD3 post fixation staining) in the top (dark red and green) and bottom (light red and green) halves of the melanocyte compartment in the chips with low melanocyte density. Statistical analysis represents the comparison of total immune cells between chips perfused with PBMCs and chips perfused with activated cells (*$P < 0.1$, $n = 4$ for NT and $n = 12$ for Act). **f** Quantification of the cytokines IL-2, IL-10, IL-6, IFN-γ, TNF-α and granzyme B on the effluents of chips after 24 h of perfusion. Statistical analysis represents the comparison between chips with low melanocyte density perfused with PBMCs and chips perfused with activated cells (***$P < 0.01$, $n = 3$–5). Bars represent averages ± SEM. Statistical analysis is detailed in Supplementary data 1, including detailed information of the number of biological replicates per group, exact $p$ values and individual data points.

a reduction on CD69 expression in CD8a positive cells, statistically significant for both concentrations (Fig. 5c), a reduction of all measured cytokines in the chip (Fig. 5d) and of all except for IL-2 in the plate controls (Fig. 5e). The IL-2 increase in chips caused by T cell activation is slightly reverted in the high CsA condition on the chip but not on the plate controls. Most importantly, these effects correlate with CsA concentration dependent reduction in immune cell recruitment. Treatment with high concentrations of CsA reversed the infiltration of immune

cells by 2-fold relative to activated T cells and 1.8-fold relative to low CsA condition (Fig. 5f, g).

CsA was selected not only for its immunosuppressive properties, but also for its effects on improving uveitis symptoms[41]. CsA has a paradoxal effect on T cells since it is able to suppress T cell activation and proliferation at high doses but, at lower doses, it is reported to have an immunomodulatory effect with potential application for immuno-oncology therapies[42]. The mechanism that supports the effect of this drug for treating uveitis is that

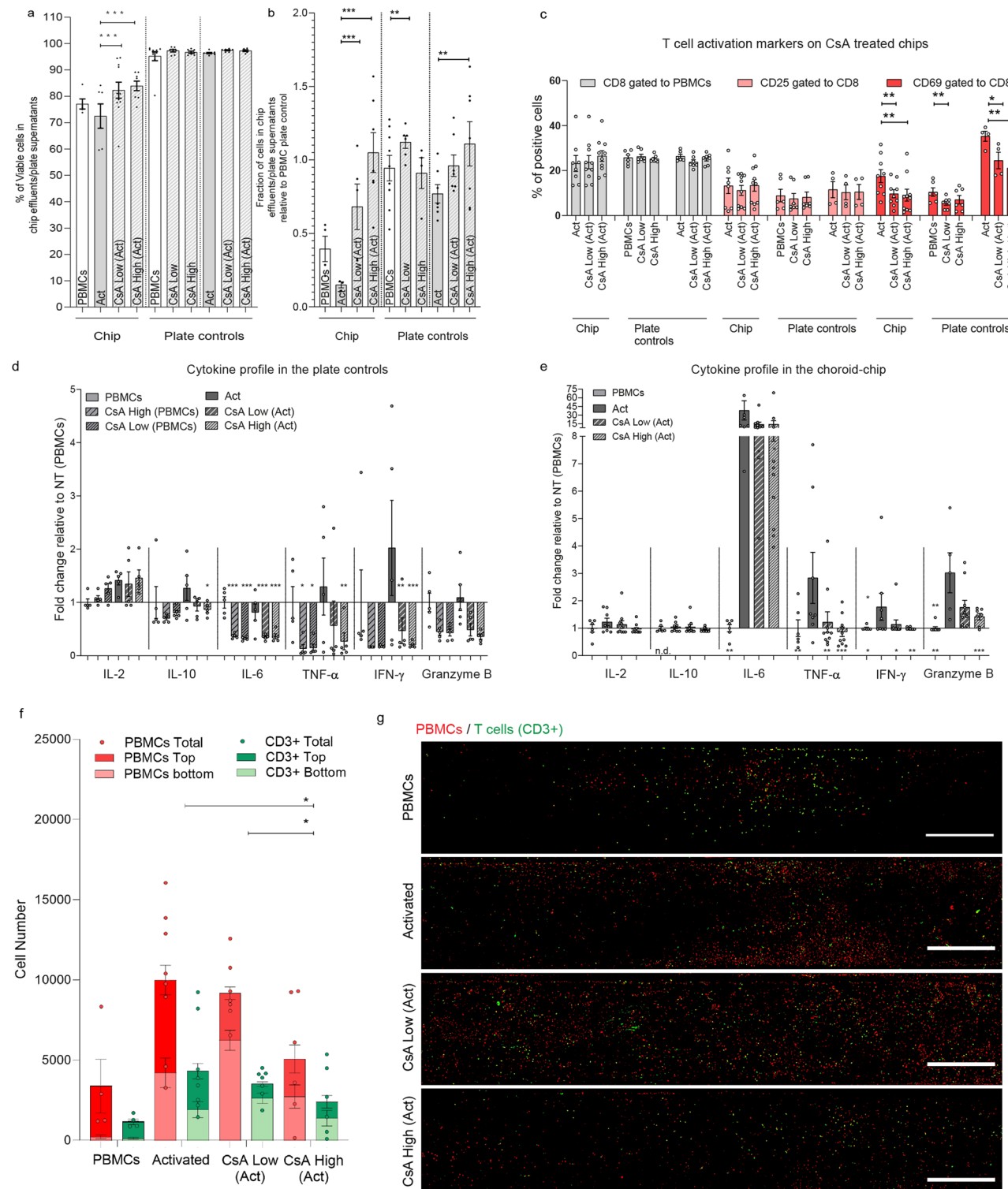

uveitis is characterized by T cell activation and infiltration into the choroidal tissue[8,43], as well as other inflammatory cells including B cells and macrophages[34,35]. As mentioned above, IL-6 is increased in posterior uveitis[37] and identified as a potential therapeutic target[44]. Our data show a similar biological response when the chip is treated with activated cells covered by cytokine production (IL-6 increase) and immune cell recruitment to the choroid. Taking a closer look into the correlation of cytokine secretion in the CoCs and the immune cell recruitment, CsA effectively inhibits IL-2 production in vivo and in vitro[45,46], as

observed for the high CsA concentration in the chip but not in the plate control. IL-2 is required to maintain the activation state of T cells[47]. Activation of T cells only led to an increase of IL-6 in the CoCs, but not in the plate control. In addition to melanocytes and RPE[35], monocytes can also produce IL-6, promoting the proliferation and migration of T cells in a concerted way[40]. CsA is able to directly reduce IL-6 production by monocytes[48], in agreement with the 10-fold reduction of IL-6 levels in CsA treated chips (Fig. 5d). The monocyte role was not explored in depth in the present work. However, monocyte migration towards uveal

**Fig. 5 Effect of the immunosuppressor drug Cyclosporine (CsA) on the Immuno-competent CoC with low melanocyte density after 24 h of perfusion or cell culture (plate controls) respectively, in PBMCs, activated PBMCs (Act), treated and non-treated with high (500 ng/mL) and low (100 ng/mL) CsA concentrations. a** Viability of the immune cells collected in the chip effluents and plate control supernatants ($n = 3$–10, the dots represent each data point). **b** Fraction of cells recovered in in the chip effluents and plate control supernatants relative to the number of recovers cells in the PBMC control ($n = 3$–10). **c** Flow cytometry quantification of CD8 fraction and the T cell activation markers CD25 and CD69 in the CD8 fraction in activated PBMCs (Act) with and without CsA treatment ($n = 4$–10). **d** Cytokine profile in the choroid chip in activated PBMCs (Act) with and without CsA treatment relative to PBMCs ($n = 6$–12). **e** Cytokine profile in plate in PBMCs and activated PBMCs (Act) with and without CsA treatment relative to PBMCs ($n = 5$). **f** Quantification of immune cell recruitment by image analysis of the whole chip, distinguishing PBMCs (cell tracker labeled prior perfusion) and T cells (CD3 post fixation staining) in the top (dark red and green) and bottom (light red and green) halves of the melanocyte compartment chips with low melanocyte density Statistical analysis represents the comparison of total immune cells between chips perfused with PBMCs and chips perfused with activated cells (*$P < 0.1$, $n = 4$–8). **g** Representative images of the melanocyte bottom compartment containing labeled PBMCs (Cell tracker, red) and T cells (CD3, green) represented as maximum intensity projections for each of the analyzed condition. Scale bar = 500 μm. Bars represent Averages ± SEM. Statistical analysis represents the comparison of the CsA treatment in activated cells with the same condition not exposed to CsA (*$P < 0.1$, **$P < 0.05$, ***$P < 0.01$), unless otherwise labeled. Statistical analysis is detailed in Supplementary data 1, including detailed information of the number of biological replicates per group, exact $p$ values and individual data points.

---

melanocytes after being in contact with conditioned medium from activated T cells that contains increased IFN-γ and TNF-α has been described in vitro[49]. T cells produce those cytokines after T cell receptor stimulation and induce monocyte production of GM-CSF. Consequent monocyte activation promotes monocyte adherence, migration, chemotaxis[50]. Our data also show elevated IFN-γ and TNF-α in the CoCs perfused with activated T cells, which might suggest a T cell dependent monocyte activation and migration, represented by the CD3 negative PBMCs population, which was reduced by more than 2-fold with high CsA concentrations (Fig. 5f). Interestingly, in the chips perfused with activated T cells, only 50% of the migrated cells were CD3 positive, independently of their depth of penetration into the melanocyte compartment. For chips perfused with PBMCs, most of the cells were in the upper half of the compartment, near the endothelial cells (Fig. 5f). This was not reverted by CsA treatment suggesting that CsA treatment affected the number of cells that infiltrate the melanocyte compartment, but not the spatial distribution of immune cell types. To observe the effect of activation on T cell proliferation and possibly a difference in the spatial distribution of immune cell types, the evaluation period would need to be increased because 24 h might be insufficient. CsA was also reported to reduce T-cell trans-endothelial migration[51]. That effect is reflected by the CsA-induced reduction of the number of T cells by 1.2- and 1.8-fold for low and high CsA, respectively, relative to the activated T cells.

**Bi-specific T cell engagers (TCBs).** Bi-specific T cell engagers (TCBs) have been investigated for immuno-oncology therapies for more than 30 years. However, there are still unresolved potential safety liabilities for this mode of action revealed in toxicity studies with non-human primates and in clinical studies[52]. TCBs are antibodies that target a constant-component of the T cell/CD3 complex and a tumor-associated antigen (TAA). The relative abundance of T cells in blood, and their cytotoxicity and proliferation capacity upon activation make T cells an effector cell of choice. To test the capacity of the CoC for immune safety assessment, we tested two antibodies, labeled as A and B. Both contain the T cell receptor binding domain but lack the TAA. Both TCBs were compared with a commercially available reagent developed to activate and expand human T cells via exposure to beads containing anti CD3/CD28 antigens. The PBMC suspension was supplemented with the different T cell activators and then perfused through the CoC for 24 h as described for the CsA study. The PBMCs recovered in all effluents showed good viability (Fig. 6a). The treatment with TCB B led to a higher cell recovery than TCB A or with the activated

condition after chip perfusion and in the plate controls (Fig. 6b). Both TCBs led to a similar low increase in IL-2 as for the commercial T cell activator (Act). TCB B elicited minimal cytokine release, with some increase mainly evident for granzyme B. TCB A led to significantly higher levels of TNF-α and IFN-γ when compared to the commercial activator, while this latter showed higher levels for IL-10, IL-6 and granzyme B. This is particularly interesting since T cell secretion of TNF-α and IFN-γ leads to monocyte migration towards uveal melanocytes[50], as discussed above. This could lead to an untargeted, undesired and possibly detrimental immune response. Our data show that TCB B is leading to similar T cell recruitment to the melanocyte compartment as TCB A and the activated condition but to less recruitment of CD3 negative cells. These observations and the higher presence of cells in the chip effluent in TCB B treated chips support its higher efficiency on T cell-specific recruitment while producing lower levels of pro-inflammatory cytokines.

## Conclusion

Altogether, this microphysiological CoC model recapitulates key features of the human choroid such as cellular composition, pigmentation, tissue-specific endothelial barriers as well as cytokine secretion and responds to immunomodulatory strategies, both to immunosuppressors and immunoactivators. Within this study we built a tool (i) for probing mechanisms of uveitic side effects of immune-related therapeutics for cancer treatment as demonstrated with the TCB data (Fig. 6) as well as (ii) for screening of uveitis treatments in a model of the inflammatory choroid, artificially obtained by systemic T cell activation, displaying the main uveitis features on the chip. Thereby, although not capturing the entire universe of events, the main molecular events of drug induced uveitis (i) inflammation and (ii) immune cell infiltration into the choroid could be mimicked; similarly, their reduction upon CsA treatment, a classical immunosuppressant used for uveitis treatment, could be recapitulated. Since drug-induced uveitis is idiosyncratic, a drug, e.g., ipilimumab, would not necessarily trigger such effects in a model. The CoC was also sensitive to detect differences in the immunological responses to two TCBs providing hints on their potential safety profile. The data presented here sustain the relevance of using an advanced cell culture system integrating several cell types involved in the immune response of a specific tissue to support pharmaceutical compound ranking and decision making. The immune response in the CoC was shown to be sensitive to the mechanisms of T cell activation and suppression, key aspects for a human-relevant healthy and diseased in vitro model. Furthermore, the processes established for building the CoC model can potentially be translated to other tissues where the stromal,

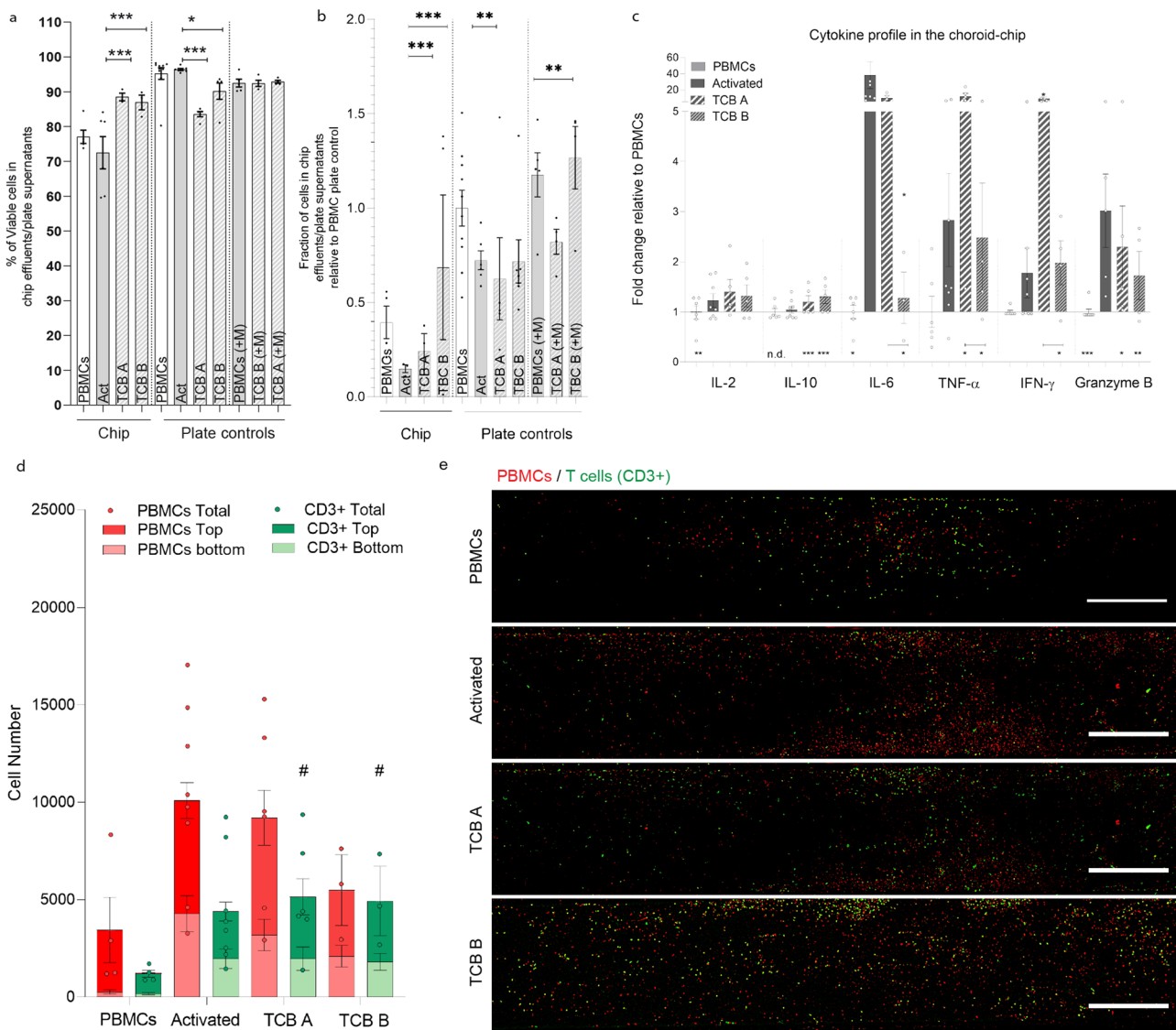

**Fig. 6 Effect of the TCBs on the Immuno-competent Choroid-chip with low melanocyte density after 24 h of perfusion or cell culture (plate controls) respectively, in PBMCs treated and non-treated with TCB A or TCB B (5 μg/mL). a** Viability of the immune cells collected in the chip effluents and plate control supernatants ($n = 3$–10, the dots represent each data point). **b** Fraction of cells recovered in the chip effluents and plate control supernatants relative to the number of recovered cells in the PBMC control ($n = 3$–10). **c** Cytokine profile in the TCB A and TCB B treated choroid chips relative to PBMCs ($n = 6$–12). **d** Quantification of immune cell recruitment by image analysis of the whole chip, distinguishing PBMCs (cell tracker labeled prior perfusion) and T cells (CD3 post fixation staining) in the top (dark red and green) and bottom (light red and green) halves of the melanocyte compartment chips with low melanocyte density ($n = 3$–9). **e** Representative images of the melanocyte compartment containing labeled PBMCs (Cell tracker, red) and T cells (CD3, green) represented as maximum intensity projections for each of the analyzed condition. Scale bar = 500 μm. Bars represent Averages ± SEM. Statistical analysis represents the comparison of the TCBs treatment in PBMCs with cells activated with antiCD3/CD28 beads (*$P < 0.1$, **$P < 0.05$, ***$P < 0.01$), unless labeled otherwise, and the comparison of the total and bottom compartments of the TCBs treatment in PBMCs with non-treated PBMCs (#$P < 0.1$). Statistical analysis is detailed in Supplementary data 1, including detailed information of the number of biological replicates per group, exact p values and individual data points.

endothelial and epithelial cells are key elements of mechanisms of disease and/or toxicity and/or expanded to test multiple donors for exploratory mechanistic studies on patient-specific idiosyncrasies.This could help building a toolbox of in vitro models recapitulating aspects of the immune system to study immune-oncology therapies such as checkpoint inhibitors, TCBs and CAR-T cells.

## Materials and methods
**Microfluidic chip fabrication by UV-lithography and replica molding.** The CoC is based on a tailored five-layered microfluidic platform comprising three microstructured polydimethylsiloxane (PDMS; Sylgard 184, Dow Corning, USA) layers,

which are separated by two isoporous, semipermeable PET membranes (Fig. 1a). 100 μm high channel structures in the PDMS pieces were generated using two differently patterned master wafers fabricated via photolithographic processes[53]. The PDMS elastomer base and curing agent were mixed in a 10:1 (w/w) ratio and desiccated to remove air bubbles. Two different replica molding approaches were conducted as described previously[53]. Standard molding was used to obtain 3 mm and 0.5 mm thick PDMS pieces with channel structures for the RPE and Melanocyte layers, respectively. Curing was achieved at 80 °C for 4 h. The endothelial layer was fabricated using exclusion molding to achieve a 0.1 mm thin layer with through hole channel structures. Layers were cured at 80 °C for 1 h. Commercially available semipermeable polyethylene terephthalate (PET) membranes ($r_P = 3$ μm; $\rho_P = 8 \times 10^5$ pores per cm$^2$; TRAKETCH® PET 3.0 p S210 × 300, SABEU GmbH & Co. KG, Northeim, Germany) were functionalized by a plasma-enhanced, chemical vapor deposition (PECVD) process as described previously[53]. Microfluidic chips

were assembled in four consecutive bonding steps: (i) RPE layer to 1st membrane, (ii) RPE layer-membrane assembly to endothelial layer, (iii) RPE-membrane-endothelial assembly to 2nd membrane, and (iv) the entire assembly to the melanocyte layer. Before bonding, all parts were cleaned thoroughly using isopropanol and blow-dried with a nitrogen pistol. Dust was removed using standard adhesive tape. In all steps, bonding was achieved by oxygen plasma activation (50 W, $0.2 \, cm^3m^{-1}$ $O_2$; Diener Zepto, Diener electronic GmbH + Co. KG, Ebhausen, Germany) for 15 s. To enhance bonding strength, assembled parts were placed at 80 °C after each step for at least 10 min and overnight after assembly of the entire chip.

Prior to cell injection, all chips were $O_2$-plasma sterilized (50 W, $0.2 \, cm^3m^{-1}$ $O_2$) for 5 min. Afterwards, channels were filled with Dulbecco's phosphate-buffered saline without $MgCl_2$ and $CaCl_2$ (PBS; Sigma-Aldrich Chemie GmbH, Steinheim, Germany) and centrifuged under sterile conditions for 3 min at $200 \times g$ to remove residual air from the systems.

**Cell isolation and culture.** All research was carried out in accordance with the rules for investigation of human subjects as defined in the Declaration of Helsinki. Patients gave a written agreement according to the permission of the Landesärztekammer BadenWürttemberg (IRB#: F-2012-078 & F-2020-166; for normal skin from elective surgeries) for the collection of primary melanocytes and microvascular endothelial cells and with the ethical Committee of the Eberhard Karls University Tübingen (Nr. 678/2017BO2, for iPSC derived retinal pigmented epithelial cells and Nr. 495/2018-BO02 for the isolation of PBMCs from whole blood). All primary mature melanocytes and microvascular endothelial cells were isolated from biopsies that were taken from female, pre-obese donors (BMI 25.0–29.9, as per the WHO classification), aged 25–65. Human PBMCs were isolated from healthy donors (BMI < 25.0, as per the WHO classification), aged 25–35.

Retinal pigmented epithelial (RPE) cells were obtained from human induced pluripotent stem cells as described previously[54]. RPE cells were maintained in B27-retinal-differentiation-medium (BRDM) consisting of DMEM/F12 and DMEM, high glucose (1:1) with 2% B27 without vitamin A, 1% non-essential aminoacids and 1% antibiotic antimycotic solution (all from ThermoFisher Scientific, USA). Cell culture media was changed daily and cells were used up to passage three. Passaging was performed using Accumax at 37 °C and 5% $CO_2$ for 10–30 min, depending on the adherence and passage. Cells were seeded at 75,000 cells per $cm^2$ in BRDM supplemented with 10% FCS, 20 ng/mL of EGF (Cell Guidance Systems), 20 ng/mL of FGF2 (Cell Guidance Systems), 10 µM of ROCK-inhibitor Y-27632 (Selleck Chem). After 24 h the medium was changed to unsupplemented BRDM and changed daily thereafter. Passaging was never performed earlier than every 4 weeks to assure a pigmented phenotype.

Primary microvascular human endothelial cells (MVECs) and melanocytes were isolated from adult human skin tissue of plastic surgeries received from Dr. Ulrich E. Ziegler (Klinik Charlottenhaus, Stuttgart, Germany). Visible blood vessels and connective tissue were removed and the skin biopsies were cut into 8 $cm^2$ pieces and further cut into strips. Tissues were incubated with a 2 U/ml dispase solution (Serva Electrophoresis, Heidelberg, Germany) in PBS overnight at 4 °C. Samples were then washed in PBS and epidermis and dermis were separated for further melanocyte and microvascular endothelial cell isolation, respectively. Human melanocytes were isolated using 0.05% trypsin/EDTA for 10 min at 37 °C. Reaction was stopped with 5% FCS and cells were collected through a 100 mm mesh. After centrifugation at $200 \times g$ for 5 min, cells were re-suspended in melanocyte growth media and seeded to a T75 cell culture flask. Isolation of MVECs from the dermis was achieved by incubation of dermis strips in 0.05% trypsin/EDTA for 40 min at 37 °C, followed by washing with PBS and mechanical collection of cells using a cell scraper. Cells were stripped through a 100 mm mesh and then centrifuged at $200 \times g$ for 5 min. Cells were cultured in endothelial cell growth media in a T75 cell culture flasks. Both melanocytes and endothelial cells were cultured with gentamicin for the first 10 days of culture and used in passage 2 or 3 for this study.

Peripheral blood mononuclear cells (PBMCs) were isolated from whole blood using the MACSprep PBMC Isolation kit (Milteny Biotec), according with the manufacturer instructions. Blood was collected up to 1 h before isolation. After isolation, PBMCs were labeled with CellTracker™ Deep Red Dye (Thermofisher Scientific) at 1 µM in X-vivo medium (Biozym) for 45 min at 37 °C and protected from light. The cell concentration and viability were monitored using the Guava® ViaCount™ Reagent (Luminex) by flow cytometry using a Guava EasyCyte HT (Guava, Luminex) according to the manufacturer instructions.

**Cell seeding on the choroid-chip.** Prior to cell injection, the RPE channel was coated with 50 µl of 50 µg/mL laminin in DMEM/F12 and DMEM, high glucose (1:1) for 2 h at 37 °C, 95% Humidity, 5% $CO_2$ (will be referred to as "incubation conditions" in the following). Afterwards, the endothelial channel was coated using 50 µl of 30 µg/mL of fibronectin (Sigma-Aldrich) and 100 µg/mL collagen type I[55] in PBS incubated for 1 h.

RPE cells were passaged as described above and seeded in the top channel using a filtration method. Briefly, 40 µL of cell suspension at a concentration of $1.0 \times 10^6$ cells/mL was added to the RPE channel while the outlet of the same channel remained closed, as well as the inlet of the endothelial channel on the side of cell seeding and both inlet and outlet of the stromal channel. This forced the medium

to flow through the membrane and the cells to block the pores until the full length of the membrane was completely covered with cells. The cells were left to adhere in the incubator for 1 h. After incubation, filtered pipet tips, with 50 µl of supplemented BRDM media each, were added to the in- and outlet of the RPE channel, while the other two channels were completely plugged. The medium was replaced by unsupplemented BRDM for additional 24 h prior to MVEC seeding.

MVEC seeding was conducted after RPE seeding and similar to the RPE seeding method with minor adaptations: The RPE channel in- and outlet were closed off completely while the stromal channel inlet was closed and its outlet open. Thirty µL of cell suspension (concentration of $1.0 \times 10^6$ cells/mL) was injected into the endothelial cell inlet. The chip was immediately flipped and placed for 15 min in an incubator, allowing cells to attach to the top membrane. Afterwards, the chip was flipped back to the upright position and placed for an additional hour in the incubator. Filter pipet tips filled with 100 µL ECGM media each were added to the in- and outlet of the endothelial channel for 24 h of static culture. RPE and stromal channels were completely closed off.

Melanocytes, embedded into a dextran-CD hydrogel, supplemented with RGD peptide (Cellendes; Catalog no. G93-1), were injected into the stromal channel of the chip, 24 h after MVEC seeding. The hydrogel was prepared according to manufacturer instructions. The melanocytes were passaged as described above and the average cell diameter assessed using ImageJ (Supplementary material 2). By calculating the proportion of the cell volume to the total stromal channel volume, high and low melanocyte concentration were achieved corresponding to the choroid of a white human (5%) and of a rhesus macaques (50%) as described in the literature[28]. The total volume of the stromal channel corresponds to 1.5 µL and a volume of 5 µL was used for cell seeding and an expansion factor of 3.68 was used to account for the melanocyte morphology change while spreading. The calculation of cell volume was performed prior to each seeding to factor in donor and passage variability on cell size. Prior to melanocyte seeding, the chip was connected to a syringe pump to actuate a media perfusion through the endothelial channel at a flow rate of 5 µL/h with positive pressure. The cell-hydrogel suspension was injected into the stromal channel, which was subsequently closed. Cells were then placed in the incubator. Perfusion rate was ramped up to 20 µL/h (after 1 h) and finally to 40 µL/h (after 24 h); the perfusion rate maintained throughout the entire experiment. Chips were cultured for two weeks before perfusing PBMCs. For the immune cell recruitment experiments in acellular chips, CCL19/MIP-3β (R&D Systems) was added to the hydrogel for a final concentration of 250 ng/mL. Plate controls were performed for the cytokine measure in the presence and absence of PBMCs. For that, melanocytes were seeded in 24-well plates at the density of 10.000 cells/$cm^2$ and kept in culture in parallel with the chip culture until the immune cell perfusion.

**Immune cell perfusion and treatment.** Isolated and pre-labeled PBMCs were cultured at a concentration of $1.0 \times 10^6$ cells/mL. Plate controls were performed for the flow cytometry and for the cytokine measurements in all treated and non-treated conditions in static 24-well cell culture plates. The total volume for each chip perfused and the plate controls was kept equally at 960 µL, assuring the same cell concentration and treatment concentration. Chips were cultured in an incubator that allowed for chip handling under sterile conditions (Incubator FlowBox™, ALS, Jena, Germany). For PBMC perfusion through the endothelial channel, the outlet of the channel was equipped with a reservoir holding the PBMC cell suspension, while the pump was changed to the withdraw mode at 40 µL/h, maintaining the flow direction. After 24 h, the chip effluents and plate control supernatants were collected for further analysis and the chips were fixed for immunofluorescent labeling.

T cell activation was achieved by adding 5 µL of anti-CD3/CD28 antigens (TransAct, Milteny Biotec) to the $1.0 \times 10^6$ cell suspension, being present for the 24 h of perfusion in the chip or static in the plate controls.

Cyclosporin (CsA) treatment was performed for the 24 h of perfusion using two concentrations, which were selected to mimic low and high plasma concentrations of 100 and 500 ng/mL[42,56] and to evaluate a dose response of its effects on chips perfused with activated T cells. Therefore, CsA treatment on chip was performed only in combination with the T cell activation. PBMCs were also treated with CsA in the plate controls to confirm that CsA treatment is not affecting the immune cells viability and cytokine production.

T cell bispecific antibody (TCB) treatment was performed for the 24 h of perfusion. Both TCB A and TCB B were added at the concentration of 5 µg/mL to PBMCs.

**Evaluation of immune cells in chip effluents and plate control supernatants.** The effluents of the immune cell perfusion were collected and centrifuged at $200 \times g$ for 5 min. The cell pellet was re-suspended in 950 µL of PBS for evaluation of cell number and viability using the Guava® ViaCount™ Reagent (Luminex) by flow cytometry (Guava, Luminex), according with the manufacturer instructions, and for flow cytometry analysis. The supernatant was centrifuged for 10 min at 10.000 × g at 4 °C to remove cell debris and stored at −80 °C for further analysis of the cytokine profile.

For flow cytometry analysis cells were washed and labeled with anti-CD8a-PerPC, anti-CD25-APC and anti-CD69-FITC (Biolegend) for 20 min at 4 °C. Antibodies were used at concentrations recommended by the manufacturer. As

### Table 1 Flow cytometry antibodies and isotype controls.

| Fluorophore | Reactivity | Antigen | Clone | Lot number | Product reference |
|---|---|---|---|---|---|
| APC | Anti-human | CD25 | BC96 | B258747 | 302610 |
| FITC | Anti-human | CD69 | FN50 | B224681 | 310904 |
| PerCP | Anti-human | CD8a | HIT8a | B241517 | 300922 |
| FITC | Mouse | IgG1,κ | MOPC-21 | B199152 | 400107 |
| APC | Mouse | IgG1,κ | MOPC-21 | B257953 | 400119 |
| PerCP | Mouse | IgG1,κ | MOPC-21 | B226117 | 400147 |

buffer solution and for the washing step PBS containing 2 mM of EDTA and 0.1% of BSA was used. Autofluorescence and isotype controls were performed to confirm similar levels of unspecific signal in every staining. Table 1 specifies the used antibodies and isotype controls. The analysis was performed using the Guava Incyte software. The PBMCs were gated on the FCS/SSC plot before gating for CD8a. The quantification of the fraction of CD25 and CD69 positive cells was performed for the CD8a positive population (Supplementary Fig. 4).

For the cytokine measurement, the samples were thawed and analyzed using the LegendPlex Human CD8/NK Panel (Biolegend), according with the manufacturer instructions, using a Guava EasyCyte HT (Guava, Luminex). The samples were diluted 1:100 to assure that the analyses are within the calibration curves. The analysis was performed on the LEGENDplex™ v.8.0 software.

**Immunofluorescence staining.** Chips were washed twice with 100 μL of PBS by gravity flow through the endothelial and RPE channel and then fixed with 4% ROTI®Histofix solution (Roth) for 10 min at room temperature (RT) using the same method. The chips were washed with PBS after fixation and stored until further processing in PBS at 4 °C. Blocking of unspecific binding and permeabilization was achieved by incubation with a 3% BSA and 0.1% Triton-X solution in PBS- for 30 min at RT. Chips stained for CD3 were not permeabilized to avoid losing the cell tracker label in the PBMCs. All primary and secondary antibodies, specified in Table 2, were used at dilutions of 1:50 and 1:100, respectively. Positive and negative controls were used for all antibodies as well as secondary antibody controls where applicable. The highest concentrations were selected considering the background/signal ratio and the high tissue to media ratio in the chip that is significantly lower than in tissue slices or cells cultured as monolayers in well plates. High antibody concentrations were tested and were required due to the high tissue to media ratio. Using 50 μl of antibody solution per chip, primary antibodies were incubated overnight in PBS at 4 °C and secondary antibodies for 1 h at RT. Chips were washed three times with PBS after each of the antibody incubation steps using gravity flow. Following the incubation with the secondary antibodies, chips were incubated with DAPI in 0.2% saponin solution in PBS for 45 min at RT. Chips were washed three times each with 0.2% saponin solution in PBS and PBS only, and then stored at 4 °C until imaged.

**Permeability assays.** For the evaluation of the endothelial and outer blood-retina barrier, chips were perfused with Carboxyfluorescein (0.377 kDa, Sigma, 21877) and Dextran Texas Red (70 kDa, Thermo Scientific, D1830) added to the EGCM media at concentrations of 100 μM and 14.3 μM, respectively. Compounds and concentrations were based on previous studies[31]. Image acquisition was performed in the dark using an incubator chamber that was fitted to the microscope stage, and set to 37 °C, using the inverted Leica DMi8 fluorescence microscope. Images were acquired of each channel (BF, Cy 5, FITC) every 15 sec over a period of 45 min. As a reference, an image was taken prior to perfusion with the fluorescent dye solution at time point 0 min (t0). The images were acquired in the inlet of the three channels and analyzed as described in Fig. 2d. Chips were perfused at a flow rate of 40 μL/h using negative pressure and the same flow direction. For quantification of the fluorescent intensities of the different markers, the mean gray value of each channel was analyzed for each time point using ImageJ2. The fluorescence intensities were plotted against the time and normalized to the minimum (background) and to the maximum intensity in each channel. A non-linear curve fit, agonist vs. response, was applied to each dataset to determine the EC50, i.e., the time where 50% of the intensity was achieved in each channel using Graphpad Prism software.

**Confocal image acquisition and analysis.** Whole chips were imaged using a Zeiss LSM 710 version 34ch Quasar NLO equipped with the inverted microscope platform Axio Observer.Z1. All images were acquired with an EC Plan-Neofluar 10x/ 0.30 M27 air objective. The sample was excited with three different lasers. A HeNe laser was used to excite the red cell tracker at 633 nm and emission was measured at 638–755 nm. FITC and DAPI were excited by an argon laser at 488 nm and Diode 405–30 at 405 nm, respectively. Emissions of these fluorochromes were measured at 495–630 nm and 410–501 nm, respectively. The pinhole of the red, green and blue channel was set to 90 μm, 23 μm and 17 μm, respectively. Digital gain was constantly kept at 1.0 for all channels. The gain master, that controls the analog amplification of voltage in the photon multiplier tube (PMT) and thus the sensitivity of the detector to the signal, was set individually for each channel and

### Table 2 Immunofluorescence antibodies.

| AB | Supplier, Reference | Host | Reactivity |
|---|---|---|---|
| CD-1 (PECAM-1) | Dako, M0823 | Mouse | Human |
| ZO-1 | ThermoFisher, 40-2200 | Rabbit | Human |
| TYRP1 | Abcam, 3312 | Mouse | Human |
| Ki-67 | R&D Systems, AF7617 | Sheep | Human |
| CD3-FITC | Biolegend, 300406 | Mouse | Human |
| Alexa Fluor 488 | ThermoFisher, A11008 | Goat | Rabbit |
| Alexa Fluor 546 | ThermoFisher, A11003 | Goat | Mouse |
| Alexa Fluor 546 | ThermoFisher, A21098 | Donkey | Sheep |

kept the same for all chips analyzed. For transmitted light images, in the figures referred to as bright field for simplicity reasons, the transmission photo multiplier tube (T-PMT) was used and set individually for each chip. For the quantification of cell migration, the melanocyte compartment of the chip was captured in 8 tiles. Each tile was imaged as 43 stacks from bottom ($z = 0$) to the top in 3 μm steps, covering the whole thickness of the melanocyte layer. Images were acquired using ZEN version (2.1).

The image processing workflow and the ImageJ macros used to generate the quantitative data are detailed in Supplementary Material 2. Briefly, image analysis consisted of pre-processing, segmentation and analysis/measurements and quantification of PBMCs and T cells in the stromal compartment. Pre-processing consisted on re-scaling all images in z-direction by the factor of 1.8 to obtain cubic voxels of 1.66 μm edge length. Next, the red, green and blue channel were sharpened by 3D unsharp masking, which consisted of one blurring step using the *Gaussian Blur 3D…* function with $\sigma_x = \sigma_y = \sigma_z = 5$ and subsequent subtraction of the blurred image from the input stack using the *Image Calculator*. Two additional 3D unsharp masking steps were performed on the red and green channel to further enhance the signal in respect to the background. The *Hybrid 3D Median filter* plugin was used to reduce noise in the red and green channel. Finally, closing was performed on the green channel by using the *Morphological Filters (3D)* function implemented in the *MorphoLibJ* plugin[57] with a ball of radius = 2, to fill circles resulting from surface staining of T cells.

Segmentation of the red and green channel was done using the *3D Watershed* as part of the *3D Segmentation* plugin[58]. Seeds were identified automatically from the input stack within a radius of 4. For red segmentation, fixed parameters were applied to all chips. The *seeds* threshold, defining the minimum value a local maxima needs to pass to be considered as a seed, was kept at 1000, and the *image* threshold for clustering voxels around seeds was set to 500. In green segmentation, these parameters were normalized against the maximum green intensity (max$_{green}$) of each specific chip using a constant of 0.043574883 for the seeds threshold and a constant of 0.021787442 for the image threshold. The 3D Watershed segmentation results in labeled object images. Position and volume of all red objects were measured using the *3D Centroid* and *3D Geometrical Measure* function implemented in the *3D Segmentation* plugin[58],respectively. Objects smaller than 30 voxels (corresponding to a volume of 137.22 μm$^3$) were eliminated in a size filtering step. The total number of PBMCs and PBMC counts in the bottom (0 μm–63 μm) and top (64 μm–126 μm) section of the melanocyte compartment were determined from the remaining objects. For T cell quantification, first labeled object images obtained from segmentation in the green channel were binarized in a way that all object voxels were assigned to 255 while background voxels were assigned to 0. Second, the proportional overlap of each red object with any green object voxel was measured using *3D Intensity Measure* (54). Finally, T cells were counted as those PBMCs that overlap at least with one green object voxel (meaning that the measured intensity is >0) and were grouped into top and bottom, similar to PBMCs.

**Statistics and reproducibility study design.** Every chip is considered an independent biological experiment. For each run of chips, the same condition was performed at least twice. For all graphs with a statistical analysis, the minimum sample size is 3. A minimum of 2 runs and a maximum of 12 runs was performed. The sample size criteria depended on the logistical complexity of the whole

experiment, primary cell availability, distribution of donors to assure paired conditions, availability of pumping systems and technical issues. The criteria was the following: for proof of concept data, the sample size is at least of 2; for the validation data (PBMCs vs Act) and for the proof of concept study a minimum of 4 (CsA study and TCB data). Every test was performed in parallel with a respective control (PBMC and Act). Chips were excluded and not analyzed if there would be technical issues with the connection and pumping system. Only chips with the correct effluent volume were considered. Replication was successful. Variability is within biological variability for a complex system with 4 different cell types cultured simultaneously. Chips were randomly allocated into experimental groups using a numbering system for distinguishing them. The numbering system was kept until analysis.

Bar graphs are represented as average values ± SEM. The number of independent experiments, biological replicates, is detailed in each figure caption. Statistically significant differences among two groups were analyzed using a One-way ANOVA for cell viability, cell fractions, CD69 expression and cytokine analysis and using mixed-effects model (Restricted Maximum Likelihood (REML)) for the migration analysis were PBMCs and CD3+ cells were considered dependent variables. This model was chosen instead of Two-way ANOVA due of the different number of biological experiments per condition and the results can be interpreted like repeated measures ANOVA[59]. Graphpad Prism software (version 8.2.0), with the threshold for significance set at $P < 0.1$. Those are referred in the text and exhaustively described in Supplementary Data 1.

**Reporting summary**. Further information on research design is available in the Nature Research Reporting Summary linked to this article.

## Data availability

The source data for the graphs and charts are available as Supplementary Data 1 and any remaining information can be obtained from the corresponding author upon reasonable request.

## Code availability

The code used for image analysis is fully uploaded in the Supplementary material 4.

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

## Acknowledgements

Eva Forsten, Stefanie Fuchs, Dominic Baum, Cristhian Rojas, Sandra Sturm, for chip production. Saskia van de Poel for technical support. Lisa Zeifang, Laura Laistner and Julia Roosz for stainings and image acquisition. The research was supported in part by the Ministry of Science, Research and the Arts of Baden-Württemberg (Az: 33-7542.2-501-1/30/9).

## Author contributions

P.L., S.K., M.C., A.S. designed the study; P.L. and C.P. designed and developed the chip production protocols, K.S. performed the COMSOL simulation; M.C., K.L. and K.S. performed experiments; M.C. collected and analyzed data; M.J.F. performed the image analysis; L.M. contributed to the RPE differentiation; M.W. contributed to the collection of immune cells. S.L., K.A., M.M., V.N., A.M.G., and A.S. provided conceptual advice.

## Funding

## Competing interests

M.M., V.N., A.S., A.M.G., and S.K. are employees of F. Hoffmann-La Roche Ltd. M.C., C.P., and P.L. hold a patent related to the technology presented in the manuscript (WO2020120466). The remaining authors declare no competing interests.
