## [Transparent Peer Review File · Communications Biology]

Reviewers' comments:

Reviewer #1 (Remarks to the Author):

The paper describes the development of a human immunocompetent Choroid-on-Chip (CoC). The composed chip is requiring following properties: biomimetic composition, controlled permeability and in vivo like response to inflammatory insult

Major point

1. Suggested biochip was well designed and sophisticated. However, I concern that Bruch's membrane was not considered in the composition. Bruch's membrane plays an important role in the pathophysiology of the choroid and is known to act as a very important variable in metabolite transport. In particular, the thickness of Bruch's membrane is one of the decisive factors of drugs that might affects RPE.
2. Although this paper presents very interesting and solid results, in order to know whether the in vitro chip is properly constructed, it must be verified whether a similar reaction can be obtained or compared with the in vitro model, especially uveitis.
3. The authors demonstrated immune cell recruitment into the stromal compartment, but it was not standardized parameter. To measure the quantity of responsiveness of the chip, the half maximal inhibitory concentration (IC50) needs to be suggested.

Minor point

The authors used several types of primary cells. Although very good results can be obtained for constructing a chip, I concerned the repeatability of the result using primary cell composed chip.

Reviewer #2 (Remarks to the Author):

This manuscript presented a immunocompetent Choroid-on-Chip integrating melanocytes, retinal pigmented epithelial cells and microvascular endothelial cells. The authors found that controlled immune cell could recruitment into the stromal compartments in chip. This chip could be used for studying ocular effects of biological drugs. However, the design of microfluidic device is not new. Several changes must be made to bring clarity and quality to the paper, prior to being considered for publication possibility.

1. The author declares that "The few available in vitro models of the human eye and especially the choroid are mostly 2D, integrate few cell types and lack vascularization or immune components; aspects that are all crucial for mechanisms of toxicity or disease triggering events" In some reports, these models were 3D, using more than two cell including endothelial cells and so on. (Lab Chip, 2018,18, 1539-1551, JOURNAL OF OCULAR PHARMACOLOGY AND THERAPEUTICS, Volume 36, Number 1, 2020, and References 21、 22)
2. There were no control (without cell) in barrier characterization (Fig2c、 d) .
3. Data should be added to confirm that the homing of immune cells is due to cell-cell interaction but gravity or collagen.
4. RPE cells were obtained from human induced pluripotent stem cells, how much is the differentiation efficiency on chip?
5. The effect exerted by melanocytes does not seem to be significant.
6. During continuous perfusion, the concentration of cytokine is continuously diluted. How to ensure the accuracy of the measurement results?

Reviewer 1:

Major comments:

- 1. Suggested biochip was well designed and sophisticated. However, I concern that Bruch's membrane was not considered in the composition. Bruch's membrane plays an important role in the pathophysiology of the choroid and is known to act as a very important variable in metabolite transport. In particular, the thickness of Bruch's membrane is one of the decisive factors of drugs that might affects RPE.**

We thank the reviewer for this important point. Bruch's membrane is of high importance for metabolite transport through the outer blood-retina barrier. As for any model, there are certain aspects that cannot be fully mimicked. We chose to integrate a PET membrane with a thickness of 12 μm , which is thicker than the BM (2-5 μm depending on the age) but it is a very thin membrane from the material perspective. This is needed to assure the chip stability and reproducibility. As described in the Methods section "Coating" (Lines 477, page 16), we try to mimic the ECM of the Bruch's membrane by coating the membrane with Laminin, Collagen and Fibronectin as described in the literature (DOI: [10.1016/j.preteyeres.2009.08.003](https://doi.org/10.1016/j.preteyeres.2009.08.003)). We recognize that this information is not detailed in the methods section or in the manuscript main text. We **included the following text to the manuscript:** „*The membranes were coated with Laminin, Collagen and Fibronectin, as described in detail in the Methods section, to mimic the Bruch's Membrane composition (ref DOI:j.preteyeres.2009.08)*”.

Further work on mimicking the Bruch's membrane composition and structure is of major interest but was out of the scope of this study. Here, we focused on the immunocompetency of the choroid and investigated the interplay of the different cell types of the choroid, in particular the immune cells.

- 2. Although this paper presents very interesting and solid results, in order to know whether the in vitro chip is properly constructed, it must be verified whether a similar reaction can be obtained or compared with the in vitro model, especially uveitis.**

The main aim of this manuscript is to show the possibility for building a tissue with an epithelial, endothelial, stromal and immune component that allows for finetuning the immune response to biological and chemical immunomodulators. To our knowledge, there is no standard validated in vitro model of uveitis that could be used as a reference. We consider that we included all the relevant controls to show that the model is sensitive to detect slight changes in the immune response and, in our perspective, this is only possible with a perfused cell culture system.

- 3. The authors demonstrated immune cell recruitment into the stromal compartment, but it was not standardized parameter. To measure the quantity of responsiveness of the chip, the half maximal inhibitory concentration (IC50) needs to be suggested.**

The information on the trend to increased/decreased recruitment relative to positive and negative controls is sufficient to support decision making and on designing new experimental settings to answer possible mechanistic questions. In our perspective, an IC50 approach would require a very large number of experiments to be run in parallel, it would be extremely time and cost intensive and would not provide information with key translational or clinical relevance.

Immune-related adverse events in oncology are commonly dose-independent as reviewed extensively (DOIs: [10.1177/1758835918764628](https://doi.org/10.1177/1758835918764628); [10.1159/000509081](https://doi.org/10.1159/000509081); [10.21037/atm.2016.07.10](https://doi.org/10.21037/atm.2016.07.10); [10.1002/cpt.394](https://doi.org/10.1002/cpt.394)) and as reported in several clinical trials (DOIs: [10.1056/NEJMoa1501824](https://doi.org/10.1056/NEJMoa1501824); [10.1016/S0140-6736\(14\)60958-](https://doi.org/10.1016/S0140-6736(14)60958-)

2). Given the application domain of the work here presented (immune oncology therapies) we do not recognize the toxicological value of the IC50 calculations.

Minor comments:

4. The authors used several types of primary cells. Although very good results can be obtained for constructing a chip, I concerned the repeatability of the result using primary cell composed chip.

It is always a challenge to choose between physiological relevance (*e.g.*, primary cells) and repeatability (cell lines). The repeatability of a result does not provide information on its relevance, *i.e.*, if the obtained data supports drug safety assessment and if it mimics human biology. To our knowledge, there are not yet robust cell lines or iPSC-derived melanocytes, endothelial cells and PBMCs that would justify the replacement of primary cells. We decided to use primary cells from several donors for the sake of physiological relevance and to be able to study immune response. Importantly and again to assure the best biological performance, we used fresh cells (never cryopreserved). As the reviewer states, we were able to obtain reproducible results. This highlights the reproducibility of the model while allowing to measure biological responses that are also observed in *in vivo* settings (T-cell infiltration). Moreover, the microscale footprint and necessity of only small number of cells enables the generation of a large number of independent tissue models from just one donor, allowing the testing of many different conditions circumventing the issue of inter-donor variability.

Reviewer 2:

1. The author declares that “The few available *in vitro* models of the human eye and especially the choroid are mostly 2D, integrate few cell types and lack vascularization or immune components; aspects that are all crucial for mechanisms of toxicity or disease triggering events” In some reports, these models were 3D, using more than two cell including endothelial cells and so on. (Lab Chip, 2018,18, 1539-1551, JOURNAL OF OCULAR PHARMACOLOGY AND THERAPEUTICS, Volume 36, Number 1, 2020, and References 21、 22)

We thank the reviewer comment. We intended to point out that none of the available models presents all those features together. Reference 21, display a very interesting 3D structure but is limited to HUVECs, a macrovascular cell line. Reference 22, shows BM's like properties but it represents a static culture and mimics only the RPE/EC interaction, excluding the melanocytes that are a key component of the choroid and of the choroidal inflammatory state. Bennet et al. 2018 (DOI: 10.1039/C8LC00158H) is very interesting and mimics the corneal components.

In order to clarify and limit to the choroidal *in vitro* models, **the text was edited as follows:**

*„The few available *in vitro* models of the human eye ~~and especially the choroid~~ are mostly 2D, integrate few cell types and lack vascularization or immune components“*

2. There were no control (without cell) in barrier characterization (Fig2c、 d) .

We agree with the reviewer. The data on control without cell is essential and **is now included in Figure 2C and 2D labelled as Hydrogel control.**

3. Data should be added to confirm that the homing of immune cells is due to cell-cell interaction but gravity or collagen.

The data presented in Figure 4a already answers this question. In a chip without any of the other cell types, *i.e.*, endothelial cells, RPE and melanocytes, immune cells migrate towards the hydrogel only if both

the cells are activated and the hydrogel contains CCL19, a chemoattractor. This excludes that the observed effect is gravity or hydrogel driven and highlights that the presence of melanocytes and endothelial cells plays an important role. The hydrogel used in this study was a synthetic dextran hydrogel that would also not be expected to be responsible for cell migration.

4. RPE cells were obtained from human induced pluripotent stem cells, how much is the differentiation efficiency on chip?

The RPE are differentiated off chip and only injected into the chip after differentiation. The detailed protocol and characterization of the RPE after 14 days in a chip (same PET membrane, same PDMS material and same chip production methodology) was previously published by our group (Figure 4, DOI:[10.7554/eLife.46188.001](https://doi.org/10.7554/eLife.46188.001)). This included showing polarization, VEGF secretion, the presence of other RPE markers such as PAX-6, Melanoma gp100 and MITF.

5. The effect exerted by melanocytes does not seem to be significant.

The tested hypothesis and the study design is centred on the question: Does T cell activation lead to T cell infiltration through the endothelial barrier? Is this methodology sensitive to detect differences in immune adverse effects of immunooncology therapies? The effect of the melanocytes is clear at several levels. Firstly, the chips with melanocytes and endothelial cells showed a baseline degree of cell migration towards the stromal compartment (Figure 4A and 4E, see also answer to Reviewer 1, question 3). Secondly, we observe the differences on the cytokine secretion under different melanocyte densities (Sup figure 2) and on the amount of recruited immune cells, especially on the proportion of T cells in the total of PBMCs (Sup figure 2). This is discussed in page 7 in the context of IL-6 levels, that have been shown to be increased in direct proportion to the melanocyte density. The authors decided to focus the manuscript on the T cell effects perspective and for simplicity reasons, decided to move the high melanocyte density data to the supplementary material.

6. During continuous perfusion, the concentration of cytokine is continuously diluted. How to ensure the accuracy of the measurement results?

The accuracy of the results depends on the method we used to quantify the cytokines (bead-based assay in a flow cytometer), which is independent of the perfusion system. The continuous perfusion as well as the addition of human PBMCs are part of the study design and on our perspective, key requirements to assure the most physiologically relevant in vitro setting. In the human eye, the blood flow transports cytokines, nutrients, and immune cells and removes cytokines and degradation products. This continuous perfusion at a low flow rate (40 μ L/h) assures that the cytokines remain in contact with the cells for a certain time in a very high cell to medium ratio. Hence, when compared to static dish culture, the dilution at any point in time is actually much lower. This is a key feature of microphysiological systems and one of the reasons for its success as extensively reviewed in the literature ([10.1016/j.tcb.2011.09.005](https://doi.org/10.1016/j.tcb.2011.09.005); [10.1021/acs.analchem.8b05293](https://doi.org/10.1021/acs.analchem.8b05293); [10.3390/bioengineering7030112](https://doi.org/10.3390/bioengineering7030112); [10.1016/j.tibtech.2020.11.014](https://doi.org/10.1016/j.tibtech.2020.11.014)).

Reviewers' comments:

Reviewer #1 (Remarks to the Author):

The purpose of this study can be understood as the screening tool for uveitic side effects of immune-related therapeutics for cancer treatment. I understood that the IC50 may not be necessary for the purpose of observing the occurrence of idiosyncratic side effects, not the LoC for the purpose of observing drug efficacy of target disease.

The author's argument is appropriate and valid.

However, although the working principle and composition of CoC are appropriate, there is still no evidence or example to screen for drug-induced uveitis with this CoC. Although the inflammatory response was reduced with the immunosuppressive drug CsA, the behavior of the immunological drug known to induce inflammation (ex. Ipilimumab) was not described. Any evidence of triggering immune reaction should be included.

Reviewer #2 (Remarks to the Author):

The authors revised the submitted manuscript in accordance with reviewer's comments.

Reviewer 1:

Major comments:

- 1. The purpose of this study can be understood as the screening tool for uveitic side effects of immune-related therapeutics for cancer treatment. I understood that the IC50 may not be necessary for the purpose of observing the occurrence of idiosyncratic side effects, not the LoC for the purpose of observing drug efficacy of target disease. The author's argument is appropriate and valid.**

However, although the working principle and composition of CoC are appropriate, there is still no evidence or example to screen for drug-induced uveitis with this CoC. Although the inflammatory response was reduced with the immunosuppressive drug CsA, the behavior of the immunological drug known to induce inflammation (ex. Ipilimumab) was not described. Any evidence of triggering immune reaction should be included.

The purpose of this study is (i) a tool for probing mechanisms of uveitic side effects of immune-related therapeutics for cancer treatment as demonstrated on the TCB data (Figure 6) and (ii) to allow the screening of uveitis treatments in a model of the inflammatory choroid, artificially obtained by systemic T cell activation (antiCD3/CD28 beads, commercially available), displaying the main uveitis features on the chip. In future studies the technology built up for this model could be used to expand testing with several donors and help to inform about the mechanistic insights of idiosyncratic effects in a patient-specific manner.

Drug-induced uveitis treatment is symptomatic (usually with steroids) because it is an idiosyncratic event (patient specific). That said, we can only mimic the main molecular events of drug induced uveitis, but not the entire universe of events. Those events are:

- Inflammation
- Immune cell infiltration into the choroid

We show that we can induce both events with T cell activation (antiCD3/CD28 beads, commercially available) and both were reduced with the CsA treatment, a classical immunosuppressant used for uveitis treatment. The evidence of Ipilimumab triggering immune reaction is shown by the elevated IFN- γ levels and increased immune cell infiltration towards the choroidal tissue. In the CoC this is observed (Figure 4). CsA is reversing these effects and could be used as a positive control for screening of uveitis treatments (Figure 5). Because drug-induced uveitis is idiosyncratic, a drug like ipilimumab would not necessarily trigger such effects in a model due to that idiosyncratic nature.

Uveitis accounts for around 1% of all immune related events after Ipilimumab treatment and those events can occur 1 to 12 months after exposure (DOI: [10.1097/ICU.0000000000000530](https://doi.org/10.1097/ICU.0000000000000530) and <https://www.ema.europa.eu/en/medicines/human/EPAR/yervoy#product-information-section>). Ipilimumab treatment targets CTL-4 in T cells and inhibits the dendritic cell inhibitory signals that hinder that T cells attack the cancer cells. The clinical features of Ipilimumab-associated uveitis reports are quite heterogeneous and its pathogenesis is poorly understood (DOI: [10.1080/09273948.2019.1577978](https://doi.org/10.1080/09273948.2019.1577978))

Testing Ipilimumab in the CoC model and showing an uveitis phenotype of a drug induced effect would depend on donor specific characteristics including disease state that are not identified yet. Most importantly, it would need testing of multiple donors in combination with different drugs which was beyond the scope of this study.

To clarify this, we added the following sentence was included in the introduction paragraphs:

Expanded testing with several donors would further help to inform about the mechanistic insights of idiosyncratic effects in a patient-specific manner.

To clarify the potential and limitations of applying this model for drug-induced uveitis studies, the following text was included in the conclusions section:

Within this study we built a tool (i) for probing mechanisms of uveitic side effects of immune-related therapeutics for cancer treatment as demonstrated with the TCB data (Figure 6) as well as (ii) for screening of uveitis treatments in a model of the inflammatory choroid, artificially obtained by systemic T cell activation, displaying the main uveitis features on the chip. Thereby, although not capturing the entire universe of events, the main molecular events of drug induced uveitis (i) inflammation and (ii) immune cell infiltration into the choroid could be mimicked; similarly, their reduction upon CsA treatment, a classical immunosuppressant used for uveitis treatment, could be recapitulated. Since drug-induced uveitis is idiosyncratic, a drug, e.g. ipilimumab, would not necessarily trigger such effects in a model.

(...)

and/or expanded to test multiple donors for exploratory mechanistic studies on patient-specific idiosyncrasies.

REVIEWERS' COMMENTS:

Reviewer #1 (Remarks to the Author):

Through this rebuttal, the intention and the use of the presented Coc chip were sufficiently presented. The revised manuscript was fully addressed in accordance with the review comments.